

# Identification of potential biomarkers and pivotal biological pathways for prostate cancer using bioinformatics analysis methods

Zihao He[1,2,3], Xiaolu Duan[1,2,3] and Guohua Zeng[1,2,3]

[1] Department of Urology, Minimally Invasive Surgery Center, The First Affiliated Hospital of Guangzhou Medical University, Guangzhou, China
[2] Guangzhou Institute of Urology, Guangzhou, China
[3] Guangdong Key Laboratory of Urology, Guangzhou, China

## ABSTRACT

**Background:** Prostate cancer (PCa) is a common urinary malignancy, whose molecular mechanism has not been fully elucidated. We aimed to screen for key genes and biological pathways related to PCa using bioinformatics method.
**Methods:** Differentially expressed genes (DEGs) were filtered out from the GSE103512 dataset and subjected to the gene ontology (GO) and Kyoto Encyclopedia of Genes and Genomes (KEGG) pathway analyses. The protein–protein interactions (PPI) network was constructed, following by the identification of hub genes. The results of former studies were compared with ours. The relative expression levels of hub genes were examined in The Cancer Genome Atlas (TCGA) and Oncomine public databases. The University of California Santa Cruz Xena online tools were used to study whether the expression of hub genes was correlated with the survival of PCa patients from TCGA cohorts.
**Results:** Totally, 252 (186 upregulated and 66 downregulated) DEGs were identified. GO analysis enriched mainly in "oxidation-reduction process" and "positive regulation of transcription from RNA polymerase II promoter"; KEGG pathway analysis enriched mostly in "metabolic pathways" and "protein digestion and absorption." Kallikrein-related peptidase 3, cadherin 1 (CDH1), Kallikrein-related peptidase 2 (KLK2), forkhead box A1 (FOXA1), and epithelial cell adhesion molecule (EPCAM) were identified as hub genes from the PPI network. CDH1, FOXA1, and EPCAM were validated by other relevant gene expression omnibus datasets. All hub genes were validated by both TCGA and Oncomine except KLK2. Two additional top DEGs (ABCC4 and SLPI) were found to be associated with the prognosis of PCa patients.
**Conclusions:** This study excavated the key genes and pathways in PCa, which might be biomarkers for diagnosis, prognosis, and potential therapeutic targets.

Corresponding author
Guohua Zeng,
gzgyzgh@vip.tom.com

## INTRODUCTION

According to the Cancer Statistics of 2019, prostate cancer (PCa) has emerged as the second most frequently diagnosed malignancy and is estimated to be the second leading

cause of cancer-related mortalities among American males (*Siegel, Miller & Jemal, 2019*). At present, prostate specific antigen (PSA) has been the most routinely used biomarker for the screening and monitoring of PCa (*Gandaglia et al., 2019*). However, elevated PSA does not necessarily indicate PCa and often leads to false positive results as well as overdiagnosis since it can also be seen in other benign lesions such as prostatitis or benign prostatic hyperplasia (*Inahara et al., 2006*; *Liu et al., 2019b*). Therefore, excavating reliable and effective biomarkers and learning hub genes involved in the biological process (BP) of PCa is urgently needed.

With the development of high-throughput sequencing technology and bioinformatic analysis methods, the gene expression omnibus (GEO) online public database has been widely utilized to screen out differentially expressed genes (DEGs), to study molecular signals and their relations, and to aid in constructing gene regulatory networks (*Clough & Barrett, 2016*). Up to now, by either analysis of a single dataset or integrated analysis of multiple datasets in GEO, several studies have dug out genes that exert important influence on the occurrence and progression of PCa, such as cadherin 1 (CDH1) (*Fang et al., 2017*), CDCA8 (*Zhao et al., 2017*), RPS21 (*Fan et al., 2018*), PIK3R1 (*He et al., 2018*), epithelial cell adhesion molecule (EPCAM) (*Lu & Ding, 2019*), LMNB1 (*Song et al., 2019b*), IGF2 (*Tan, Jin & Wang, 2019*), and IKZF1 (*Tong, Song & Deng, 2019*). However, the key genes detected by the above studies are largely different from each other and had little in common, and such discrepancy could be attributed to the fact that PCa is considered as a heterogenous disease as a whole (*Liu et al., 2019a*; *Thomas & Pachynski, 2018*). As a consequence, further studies in this regard is still required for the exploration and validation of key genes.

In *Brouwer-Visser et al. (2018)* studied the immune microenvironment in four solid cancer types including PCa and uploaded the corresponding gene expression profile dataset GSE103512. For the first time, the present study employed the GSE103512 dataset to screen out the DEGs between 60 PCa and seven matched normal prostate (NP) tissue, and a total of 252 DEGs were detected. The gene ontology (GO) functional annotation and Kyoto Encyclopedia of Genes and Genomes (KEGG) pathway enrichment analysis of the DEGs were performed, following by the construction of the protein–protein interaction (PPI) network which could uncover the underlying molecular mechanisms involved in the development of PCa. The cytoHubba and MCODE plugins of the Cytoscape software was applied to identify the hub genes and functional modules in the constructed PPI network, respectively. The Cancer Genome Atlas (TCGA) database, the Oncomine database, and the results reported by former studies were utilized for validation of our outcomes.

## MATERIALS AND METHODS

### Microarray data

The GSE103512 gene expression profile CEL files, which consisted of 60 PCa samples of different TNM staging and Gleason grading and seven matched NP samples (*Brouwer-Visser et al., 2018*), were downloaded from the GEO database (https://www.ncbi.nlm.nih.gov/geo/). The gene probe IDs in the CEL files were converted to their corresponding gene symbols based on the annotation information of the platform GPL13158 (Affymetrix HT HG-U133+ PM Array Plate), which contains 54,715 probes.
## Data processing and screening of DEGs

The CEL files of GSE103512 were read using the affy package of the R programing language (Ver. 3.6.0). Data preconditioning (including background adjustment, normalization, and summarization) were performed using the Robust Multi-array Average method. The limma package of R was adopted to identify the DEGs between PCa and NP samples, with a $|\log_2$ fold change$| \geq 1$ and a $P < 0.05$ was deemed to be of statistically significance (*Ritchie et al., 2015*). The overall upregulated and downregulated DEGs information were saved for further analyses.

## Functional and enrichment analyses of DEGs

The present study adopted the Database for Annotation Visualization and Integrated Discovery (DAVID; https://david.ncifcrf.gov/) to obtain GO annotation and KEGG pathway enrichment information of the DEGs identified previously (*Dennis et al., 2003*). Results with $P < 0.05$ with count number $\geq 2$ were regarded as statistically significant.

## PPI network construction and modules analyses

Protein–protein interaction analysis serves as an entry point for better interpretation of relationships between different proteins on a genome-wide scale, and might be helpful to provide novel insights into protein function explanation (*Stelzl et al., 2005*). PPI relationships of DEGs was analyzed and the corresponding PPI network was constructed by the Search Tool for the Retrieval of Interacting Genes (STRING) database online tool (http://string-db.org/) (*Von Mering et al., 2003*). All DEGs were subsequently submitted to the Cytoscape software for network visualization and hub gene identification, which was realized by the cytoHubba plugin. In addition, the MCODE plugin of Cytoscape was applied to identify potential functional modules in the PPI network with default parameters (degree cutoff $\geq 2$, node score cutoff $\geq 2$, $K$-core $\geq 2$, and max depth = 100) (*Bandettini et al., 2012*).

## Validation of hub genes

First, we performed a literature search regarding gene expression profiles of PCa in Pubmed and then extracted hub genes reported by the eligible studies, which was used to compare with those identified in the present study. Next, we examined the relative mRNA expression levels of the identified hub genes using the Gene Expression Profiling Interactive Analysis (GEPIA) and Oncomine online tools for further validation. The GEPIA (http://gepia.cancer-pku.cn/) provides comprehensive online services based on TCGA database, and we used the differential expression analysis and patient survival analysis functions in this case (*Tang et al., 2017*). The Oncomine online database (https://www.oncomine.org/) was applied to examine the mRNA expression of hub genes in both multiple cancers and their corresponding normal tissues (*Rhodes et al., 2004*).

## Survival analysis of hub genes

Provided by the University of California Santa Cruz (UCSC), the UCSC Xena online tools (https://xenabrowser.net) enables researchers to study functional genomic

**Table 1 Top 10 upregulated and downregulated DEGs between PCa and NP tissues.**

| Gene symbol | Log$_2$FC | adj.P.Val | State |
|---|---|---|---|
| NPY | −3.362711603 | 0.000789175 | Upregulated |
| TARP | −3.206862002 | 4.85E-09 | Upregulated |
| OR51E2 | −2.828448822 | 7.28E-05 | Upregulated |
| NKX3-1 | −2.528690546 | 3.58E-12 | Upregulated |
| MSMB | −2.517978536 | 5.11E-06 | Upregulated |
| ACPP | −2.474802744 | 2.29E-09 | Upregulated |
| SLC45A3 | −2.46284327 | 2.66E-08 | Upregulated |
| C15orf21 | −2.327851135 | 0.003615423 | Upregulated |
| ABCC4 | −2.261776258 | 1.46E-08 | Upregulated |
| LOC100287445 | −2.170028348 | 0.000544702 | Upregulated |
| FOLR1 | 3.005860886 | 8.83E-09 | Downregulated |
| KRT13 | 2.958317838 | 2.03E-08 | Downregulated |
| SEMG1 | 2.881097184 | 0.001633121 | Downregulated |
| SEMG2 | 2.444264593 | 0.00246384 | Downregulated |
| OLFM4 | 2.410939695 | 0.000494142 | Downregulated |
| SLPI | 2.391184301 | 3.98E-05 | Downregulated |
| SCGB1A1 | 2.324648206 | 2.40E-05 | Downregulated |
| OSR1 | 2.028908547 | 3.00E-15 | Downregulated |
| HSPB6 | 1.791949206 | 1.19E-06 | Downregulated |
| CFD | 1.71042558 | 1.42E-09 | Downregulated |

Note:
DEG, differentially expressed gene; PCa, prostate cancer; NP, normal prostate; FC, fold change; adj.P.Val, adjusted *P*-value.

datasets for correlations between genomic and/or phenotypic variables. In this case we used this tool to study whether the expression of hub genes was correlated with the survival of PCa patients from TCGA cohorts. Patients were categorized into a relatively high and low expression group respectively based on the median expression value of genes, and their overall survivals (OSs) were analyzed using the Kaplan–Meier method with a log-tank test. A $P < 0.05$ was deemed as statistically significant.

## RESULTS

### Data processing and screening of DEGs

The PCa microarray expression dataset GSE103512 contained information of mRNA expression of 60 PC samples and seven matched NP samples. A total of 19,918 official gene symbols were discerned and the gene expression matrix was constructed.
The DEGs were filtered using the limma R package (criteria: adjusted $P < 0.05$ and $|\log_2$ fold change$| \geq 1$). A total of 252 DEGs were identified between PCa and NP samples, including 186 upregulated genes and 66 downregulated genes. The top 10 upregulated and downregulated DEGs based on fold changes are listed in Table 1. The volcano plot of all the genes detected and the cluster heatmap of the 252 DEGs were collected in Fig. 1.
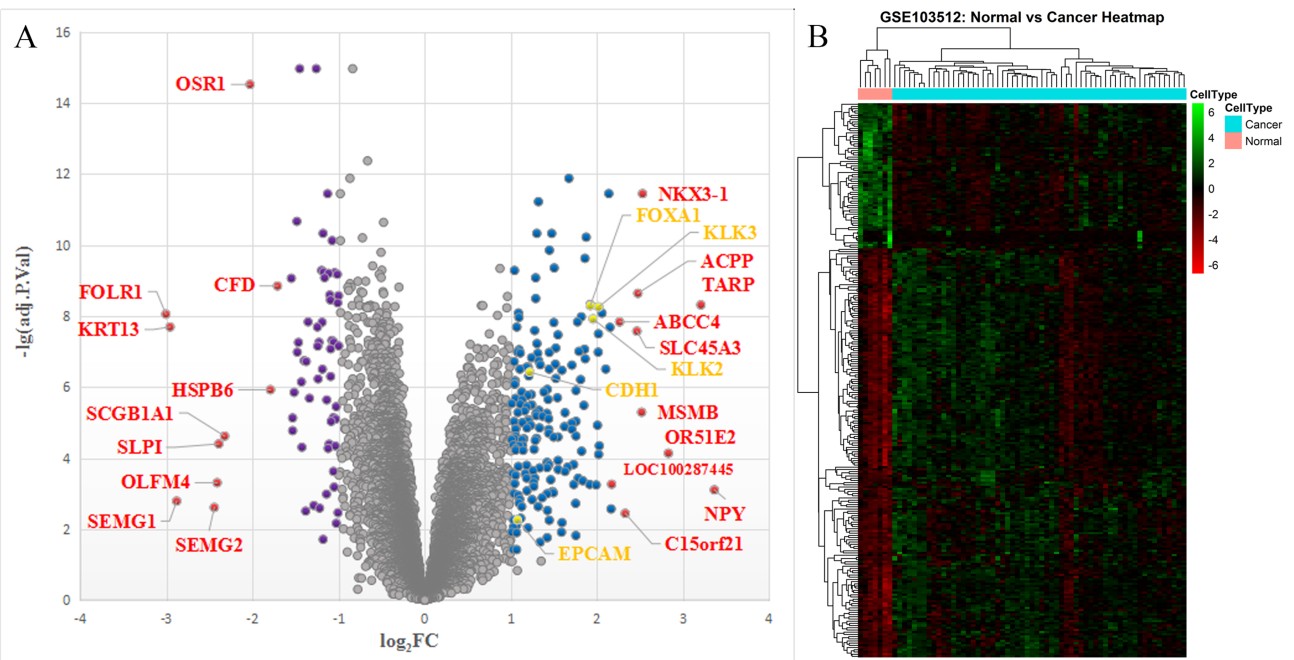

**Figure 1 The volcano plot and heatmap of the DEGs.** (A) Volcano plot: The abscissa represents $\log_2$FC and the ordinate represents $-\log_{10}$(adjusted $P$-value). The blue, purple, gray, red, and yellow dots represent significantly upregulated DEGs, significantly downregulated DEGs, insignificant DEGs, top 10 up/downregulated DEGs, and hub genes between PCa and NP tissues, respectively. (B) Cluster heatmap of the DEGs. DEG, differentially expressed gene; FC, fold change; PCa, prostate cancer; NP, normal prostate.

## Functional and pathway enrichment analyses of DEGs

Gene ontology function annotation of the identified DEGs was obtained using the DAVID database and its online analysis tool, which included the following three portions: BP, cell component (CC), and molecular function (MF). The results were deemed as statistically significant if $P < 0.05$, and the top 15 GO terms of the upregulated and downregulated DEGs are compiled in Table 2. As shown in Fig. 2A, the upregulated genes were mainly enriched in oxidation-reduction process (ontology: BP), extracellular exosome (ontology: CC), and calcium ion binding (ontology: MF). As shown in Fig. 2B, the downregulated genes were mainly enriched in positive regulation of transcription from RNA polymerase II promoter (ontology: BP), extracellular exosome (ontology: CC), and protein binding (ontology: MF).

The ensuing KEGG pathway analyses were also performed using the online analysis tool of DAVID database and the results were collected in Table 3. As shown in Fig. 2C, the upregulated DEGs enriched in five pathways: (1) metabolic pathways, (2) fatty acid (FA) metabolism, (3) arginine and proline metabolism, (4) Peroxisome-activated receptor (PPAR) signaling pathway, and (5) FA biosynthesis. As shown in Fig. 2D, the downregulated DEGs enriched in five pathways: (1) protein digestion and absorption, (2) aldosterone-regulated sodium reabsorption, (3) carbohydrate digestion and absorption, (4) mineral absorption, and (5) endocrine and other factor-regulated calcium reabsorption. The KEGG pathways bubble plot of all DEGs was presented in Fig. 3.

**Table 2 Top 15 enriched gene ontology terms of the upregulated and downregulated DEGs.**

| Category | Term | Count | P-value | State |
|---|---|---|---|---|
| BP | Oxidation-reduction process | 17 | 1.10E-04 | Upregulated |
| BP | Proteolysis | 11 | 0.01703296 | Upregulated |
| BP | Long-chain fatty-acyl-CoA biosynthetic process | 6 | 3.93E-05 | Upregulated |
| BP | Response to organic cyclic compound | 4 | 0.01023923 | Upregulated |
| BP | Response to endoplasmic reticulum stress | 4 | 0.031611372 | Upregulated |
| CC | Extracellular exosome | 74 | 2.91E-19 | Upregulated |
| CC | Integral component of membrane | 65 | 0.001627446 | Upregulated |
| CC | Plasma membrane | 56 | 7.29E-04 | Upregulated |
| CC | Membrane | 34 | 0.00188457 | Upregulated |
| CC | Extracellular space | 27 | 1.58E-04 | Upregulated |
| MF | Calcium ion binding | 13 | 0.032748281 | Upregulated |
| MF | Oxidoreductase activity | 9 | 5.26E-04 | Upregulated |
| MF | Actin binding | 9 | 0.004195153 | Upregulated |
| MF | Serine-type peptidase activity | 7 | 2.40E-05 | Upregulated |
| MF | Serine-type endopeptidase activity | 7 | 0.030442507 | Upregulated |
| BP | Positive regulation of transcription from RNA polymerase II promoter | 8 | 0.047270726 | Downregulated |
| BP | Negative regulation of transcription from RNA polymerase II promoter | 7 | 0.034624047 | Downregulated |
| BP | Negative regulation of cell growth | 4 | 0.008047189 | Downregulated |
| BP | Extracellular matrix organization | 4 | 0.028910713 | Downregulated |
| BP | Collagen fibril organization | 3 | 0.007739592 | Downregulated |
| CC | Extracellular exosome | 23 | 3.41E-05 | Downregulated |
| CC | Extracellular space | 20 | 2.27E-08 | Downregulated |
| CC | Extracellular region | 15 | 4.96E-04 | Downregulated |
| CC | Extracellular matrix | 6 | 0.002673098 | Downregulated |
| CC | Blood microparticle | 4 | 0.013152355 | Downregulated |
| MF | Protein binding | 34 | 0.021626321 | Downregulated |
| MF | Drug binding | 4 | 0.001392245 | Downregulated |
| MF | Protein dimerization activity | 4 | 0.009398722 | Downregulated |
| MF | ATPase activity | 4 | 0.016027571 | Downregulated |
| MF | Sodium:potassium-exchanging ATPase activity | 2 | 0.028658029 | Downregulated |

**Note:**
BP, biological process; CC, cell component; MF, molecular function.

## PPI network construction and modules analyses

The STRING online database was applied to analyze the 252 identified DEGs and to construct a PPI network, which consisted of 181 nodes interacting with each other via 403 edges. The results were downloaded for further analysis by Cytoscape software. According to the descending order of degree value, the top five hub genes were subsequently screened, namely Kallikrein-related peptidase 3 (KLK3), CDH1, Kallikrein-related peptidase 2 (KLK2), forkhead box A1 (FOXA1), and EPCAM, as presented in Table 4.

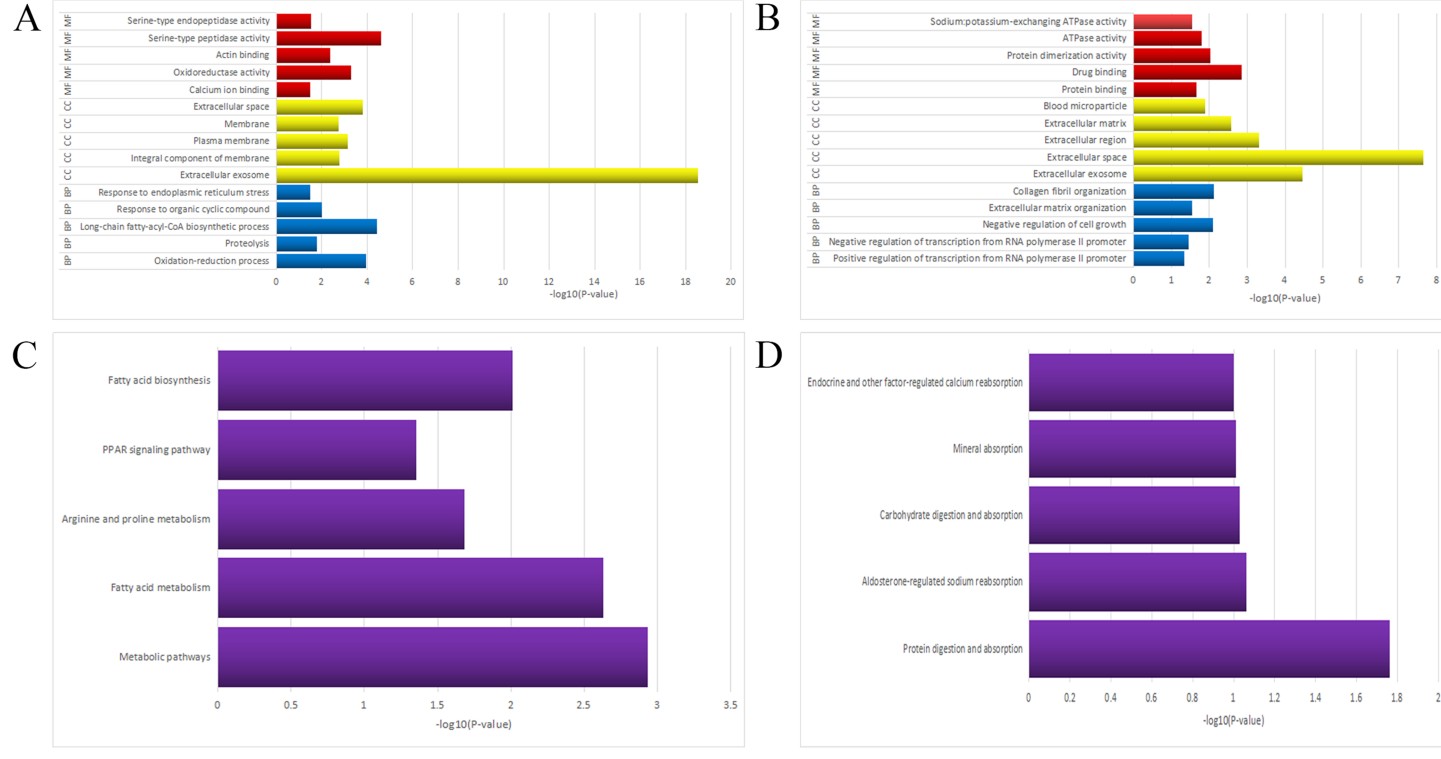

**Figure 2 The GO/KEGG enrichment analyses of the DEGs.** (A) Top 15 enriched GO terms of the upregulated DEGs. (B) Top 15 enriched GO terms of the downregulated DEGs. (C) Top five enriched pathways of the upregulated DEGs. (D) Top five enriched pathways of the downregulated DEGs. GO, Gene Ontology; KEGG, Kyoto Encyclopedia of Genes and Genomes; BP, biological process; CC, cell component; MF, molecular function.

**Table 3 Top 5 enriched pathways of the upregulated and downregulated DEGs.**

| Pathway ID | Name | Count | P-value | Genes | State |
|---|---|---|---|---|---|
| hsa01100 | Metabolic pathways | 27 | 0.001161421 | SAT1, NAMPT, SORD, GCNT2, GALNT7, GLUD1, PPT1, GGT1, ACSL1, ALDH1A3, ARG2, FASN, AMD1, ACSL3, DHCR24, UAP1, ACLY, ADI1, RDH11, FOLH1, ALOX15B, PLA2G7, PLA2G2A, ABAT, MBOAT2, SMS, DCXR | Upregulated |
| hsa01212 | Fatty acid metabolism | 5 | 0.002342524 | ACSL1, SCD, FASN, PPT1, ACSL3 | Upregulated |
| hsa00330 | Arginine and proline metabolism | 4 | 0.020729538 | SAT1, ARG2, AMD1, SMS | Upregulated |
| hsa03320 | PPAR signaling pathway | 4 | 0.043990087 | ACSL1, SCD, DBI, ACSL3 | Upregulated |
| hsa00061 | Fatty acid biosynthesis | 3 | 0.009820063 | ACSL1, FASN, ACSL3 | Upregulated |
| hsa04974 | Protein digestion and absorption | 3 | 0.017282616 | ATP1B1, ATP1A2, COL5A3 | Downregulated |
| hsa04960 | Aldosterone-regulated sodium reabsorption | 2 | 0.087044909 | ATP1B1, ATP1A2 | Downregulated |
| hsa04973 | Carbohydrate digestion and absorption | 2 | 0.093437569 | ATP1B1, ATP1A2 | Downregulated |
| hsa04978 | Mineral absorption | 2 | 0.097676003 | ATP1B1, ATP1A2 | Downregulated |
| hsa04961 | Endocrine and other factor-regulated calcium reabsorption | 2 | 0.099788246 | ATP1B1, ATP1A2 | Downregulated |

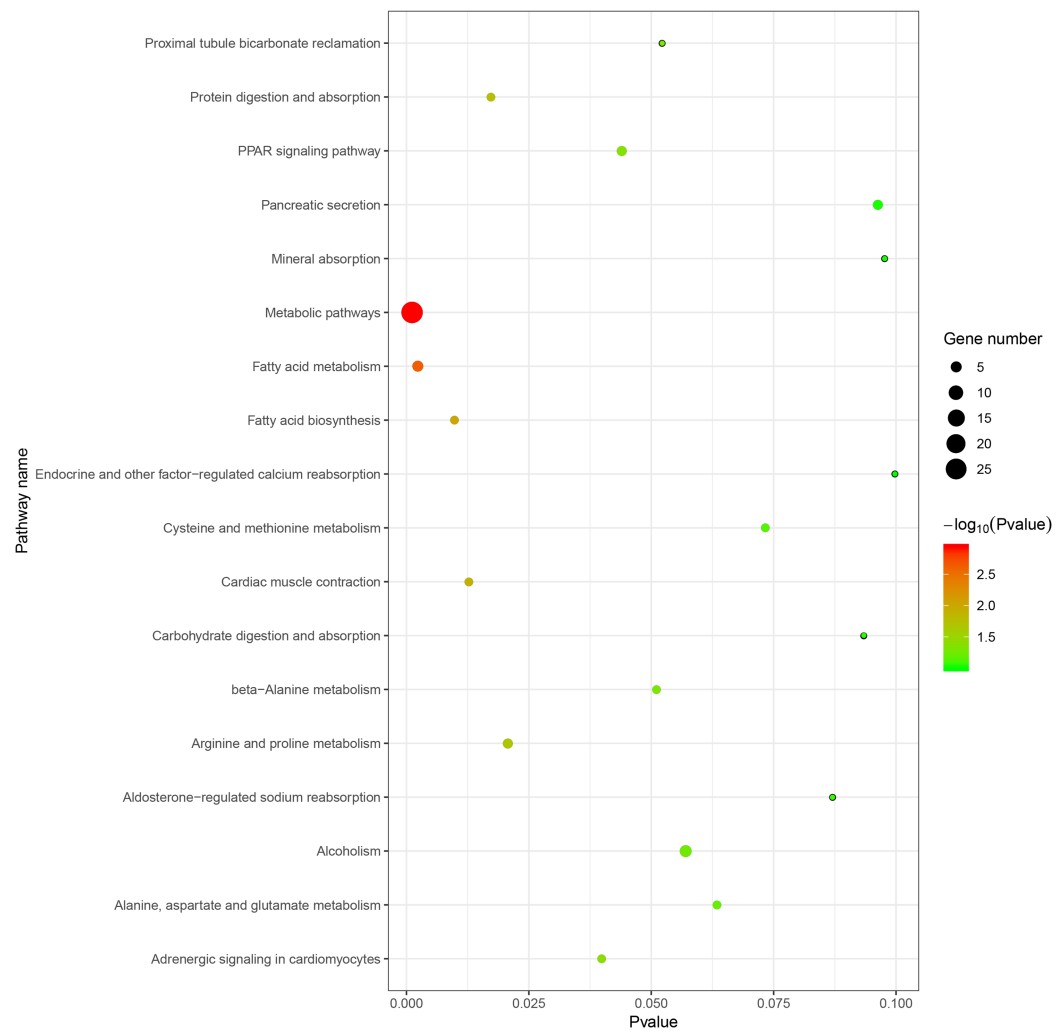

**Figure 3** **The KEGG pathways bubble plot of DEGs.**

**Table 4** **Top five hub genes in the protein–protein interaction (PPI) network.**

| Gene symbol | Gene description | Degree | Log₂FC | P-Value |
|---|---|---|---|---|
| KLK3 | Kallikrein-related peptidase 3 | 26 | 2.019308668 | 2.04E-11 |
| CDH1 | Cadherin 1 | 22 | 1.21816524 | 5.18E-09 |
| KLK2 | Kallikrein-related peptidase 2 | 21 | 1.954704556 | 5.64E-11 |
| FOXA1 | Forkhead Box A1 | 20 | 1.922009952 | 1.66E-11 |
| EPCAM | Epithelial cell adhesion molecule | 19 | 1.076626501 | 0.000999371 |

Furthermore, seven functional cluster modules were filtered from the PPI network using the Molecular Complex Detection (MCODE) plugin of the Cytoscape software, and the top two modules (Fig. 4) were selected for further KEGG pathway enrichment analyses using the DAVID database. As shown in Table 5, the results indicated that the genes in Module 1 are mainly enriched in adherens junction and Hippo signaling pathway;

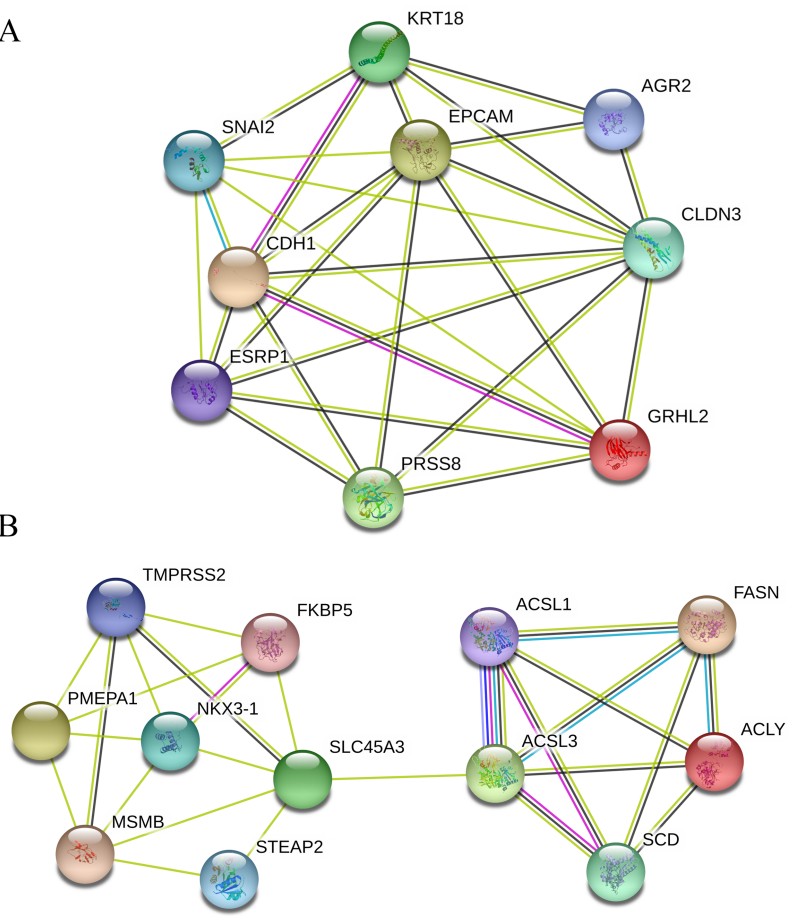

**Figure 4 Top two modules extracted from the PPI network.** (A) Module 1. (B) Module 2. Circles represent genes, lines represent interactions between gene-encoded proteins and line colors represent evidence of interactions between proteins. PPI, protein–protein interaction.

**Table 5 Pathway enrichment analysis of the genes in the top two Modules.**

| Module | Pathway | Count | *P*-value | Genes |
|---|---|---|---|---|
| Module 1 | Adherens junction | 2 | 1.0E-2 | CDH1, SNAI2 |
| | Hippo signaling pathway | 2 | 2.0E-2 | CDH1, SNAI2 |
| Module 2 | Fatty acid metabolism | 4 | 1.97E-05 | ACSL1, SCD, FASN, ACSL3 |
| | Fatty acid biosynthesis | 3 | 9.79E-05 | ACSL1, FASN, ACSL3 |
| | PPAR signaling pathway | 3 | 0.002925067 | ACSL1, SCD, ACSL3 |
| | Fatty acid degradation | 2 | 0.047850686 | ACSL1, ACSL3 |

while the genes in Module 2 are mainly enriched in FA metabolism, FA biosynthesis, PPAR signaling pathway, and FA degradation.

## Validation of hub genes expression in multiple databases

As shown in Table 6, the literature search in Pubmed/GEO regarding gene expression profiles of PCa versus NP tissues yielded 10 relevant studies, five (*Chen et al., 2012*; *Endo*

**Table 6 Comparisons between the present study and other studies using the GEO datasets.**

| Study | Dataset | Sample size (PCa, NP) | Hub genes |
|---|---|---|---|
| *Endo et al. (2009)*. | – | 3, 1 | TSPAN1, BMPR1B, **FOXA1**, STEAP1, RCAN3, S100P, LYZ, SCGB3A1, IL8, DPT |
| *Chen et al. (2012)* | – | 4, 4 | PPFIA2, PTPRT, PTPRR, PRR16, CHRM2, KRT23, CYP3A4, CYP3A7, DPYS, DUOXA1 |
| *Fang et al. (2017)* | GSE26910 | 6, 6 | **CDH1**, BMP2, NKX3-1, PPARG, PRKAR2B |
| *Zhao et al. (2017)* | GSE38241, GSE3933 | 47, 37 | CDCA8, CDCA5, UBE2C, TK1 |
| *Fan et al. (2018)* | GSE55945 | 18, 8 | RPS21, FOXO1, BIRC5, POLR2H, RPL22L1, NPM1 |
| *He et al. (2018)* | GSE46602 | 36, 14 | PIK3R1, BIRC5, ITGB4, RRM2, TOP2A, ANXA1, LPAR1, ITGB8 |
| *Lu & Ding (2019)* | GSE32448, GSE45016, GSE46602, GSE104749 | 90, 59 | **EPCAM**, TWIST1, CD38, VEGFA |
| *Song et al. (2019b)* | GSE6919, GSE6956, GSE32448, GSE32571, GSE35988, GSE46602, GSE68555, GSE69223, GSE70768, GSE88808 | 569, 402 | LMNB1, TK1, RACGAP1, ZWINT |
| *Tan, Jin & Wang (2019)* | GSE38240, GSE26910 | 14, 10 | IGF2, GATA5, F10, CFI, AGTR1, **FOXA1**, BZRAP1-AS1, KRT8 |
| *Tong, Song & Deng (2019)* | GSE26910, GSE30174, GSE46602, GSE55945, GSE69223 | 140, 53 | IKZF1, PPM1A, FBP1, SMCHD1, ALPL, CASP5, PYHIN1, DAPK1, CASP8 |
| The present study | GSE103512 | 60, 7 | KLK3, **CDH1**, KLK2, **FOXA1**, **EPCAM** |

**Note:**
Bold fonts represent overlapped hub genes among studies.

*et al., 2009*; *Fan et al., 2018*; *Fang et al., 2017*; *He et al., 2018*) of which were based on a single GEO dataset whereas the other five studies (*Lu & Ding, 2019*; *Song et al., 2019b*; *Tan, Jin & Wang, 2019*; *Tong, Song & Deng, 2019*; *Zhao et al., 2017*) were integrated bioinformatic analyses based on multiple GEO datasets. The hub genes reported by the 10 eligible studies were extracted and compared with those identified in the present study. As marked in bold fonts in Table 6, there were some overlapped findings between our studies and other studies: our identified hub genes FOXA1, CDH1, and EPCAM were seen in two (*Endo et al., 2009*; *Tan, Jin & Wang, 2019*), one (*Fang et al., 2017*), and one (*Lu & Ding, 2019*) other studies, respectively.

Next, we further validated our findings using data from TCGA. GEPIA was applied to determine the expression differences of hub genes between PCa and NP tissues. As shown in Fig. 5A, mRNA levels of KLK3, CDH1, FOXA1, and EPCAM were significantly upregulated in PCa samples compared with NP samples; but there was no significant difference in the mRNA level of KLK2 between PCa and NP samples. Besides, we investigate the prognostic values of hub genes using TCGA data and conducted survival analyses with UCSC Xena online tools. As suggested in Fig. 5B, all five hub genes had no statistically significant impact on PCa patients' OSs. Furthermore, we put the top 10 significantly upregulated and downregulated DEGs into survival analysis and found two

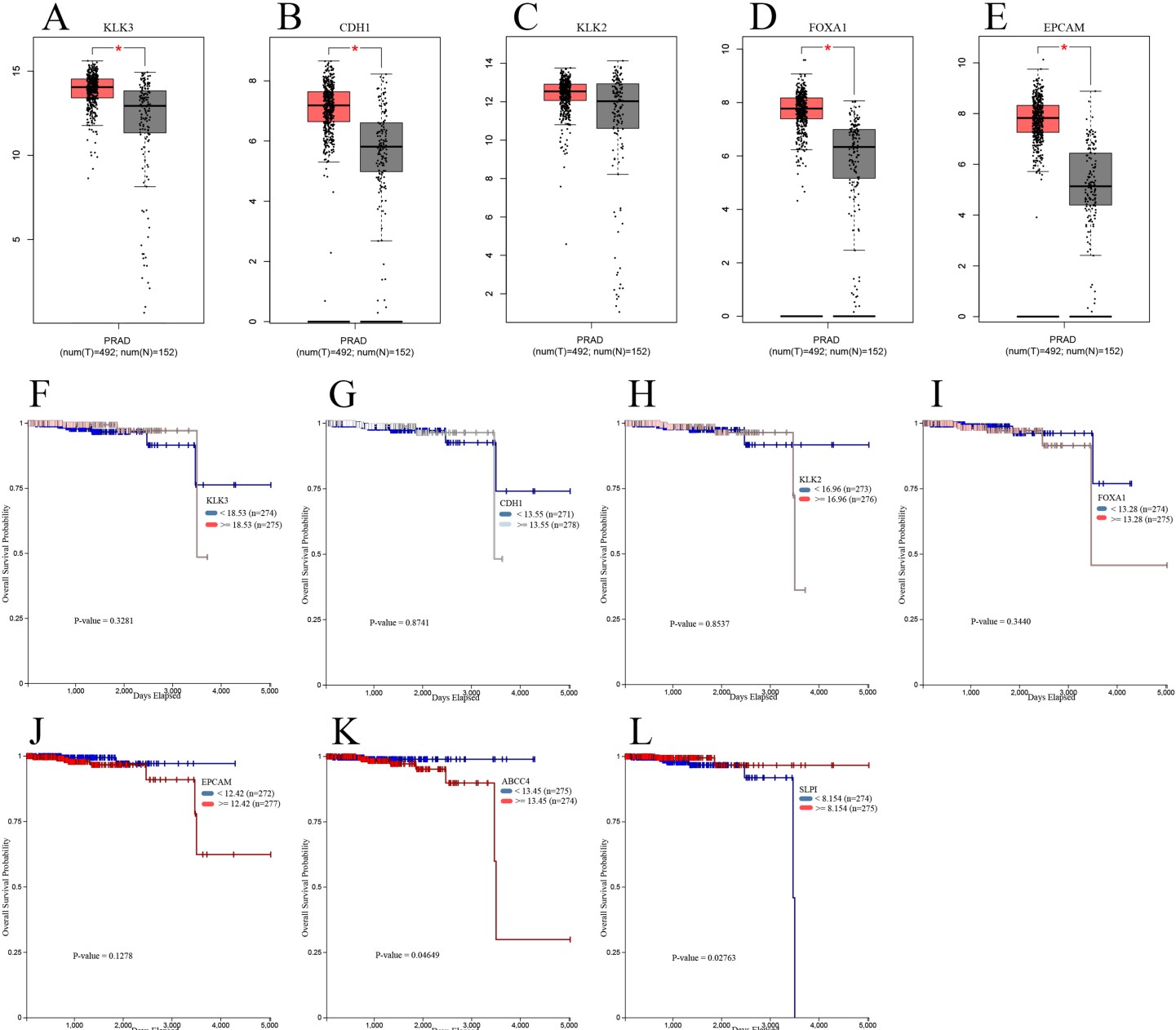

**Figure 5 Validation of hub genes using TCGA data.** Relative expression of hub genes between PCa and NP samples in the GEPIA database (the threshold was set as |log$_2$FC| = 1, P = 0.05, *P < 0.05): (A) KLK3; (B) CDH1; (C) KLK2; (D) FOXA1; (E) EPCAM. The relationship of hub genes and top DEGs with the overall survival of TCGA PCa cohorts analyzed by the UCSC Xena online tools: (F) KLK3; (G) CDH1; (H) KLK2; (I) FOXA1; (J) EPCAM; (K) ABCC4; (L) SLPI. TCGA, The Cancer Genome Atlas; GEPIA, the Gene Expression Profiling Interactive Analysis database; UCSC, University of California Santa Cruz.

genes had significant impact: the relative higher expression of ABCC4 (upregulated) as well as relative lower expression of SLPI (downregulated) were associated with poorer OS.

Furthermore, an overview of hub genes expression in multiple types of cancers revealed that all five hub genes were remarkably overexpressed in PCa tissues compared with NP tissues according to the Oncomine database (Fig. 6).

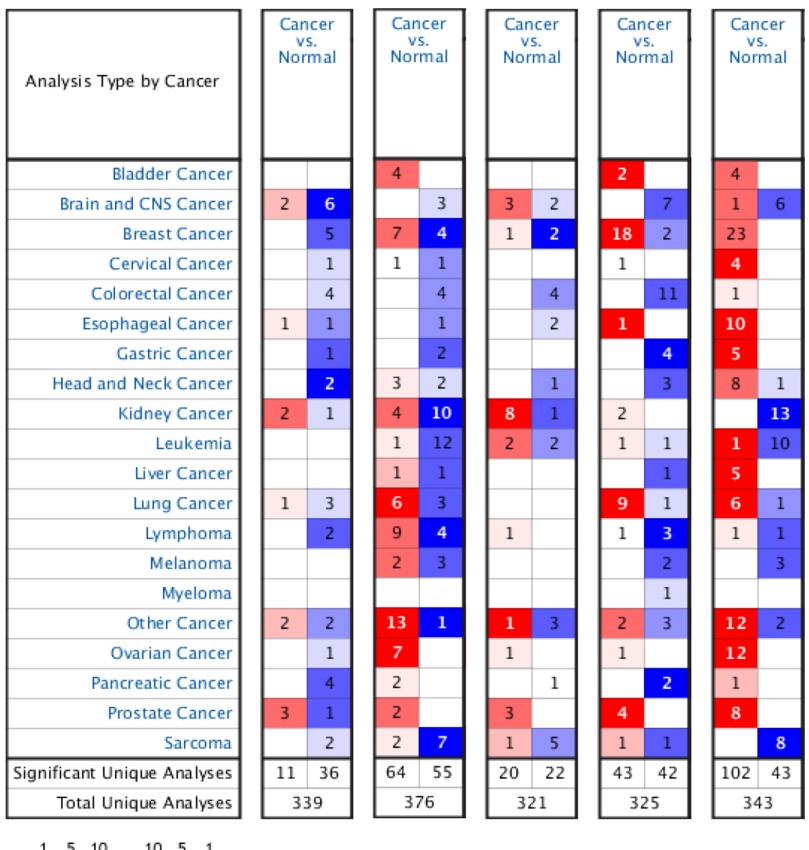

Cell color is determined by the best gene rank percentile for the analyses within the cell.

NOTE: An analysis may be counted in more than one cancer type.

**Figure 6** **An overview of mRNA levels of hub genes in various types of cancer based on Oncomine data.** The numbers in colored grids shows the counts of datasets with statistically significant mRNA overexpression (red) or low expression (blue) of genes. Grid color was determined by the best gene rank percentile for analysis within the grids. The threshold was set as gene rank percentile = All, $P$ = 0.05, and FC = 2.

## DISCUSSION

As one of the leading healthcare concerns worldwide, PCa has become one of the most prevalent malignancies of adult males with an incidence of around 0.01% (*Jemal et al., 2010*). Although numerous progress has been made to uncover the molecular mechanisms of PCa development and progression, the outcomes remain obscure and the inconsistencies among different published studies are obvious (*Barbieri et al., 2017*). The main reason for such phenomenon is generally thought to be the complexity and heterogeneity nature of PCa as a whole (*Liu et al., 2019a*; *Thomas & Pachynski, 2018*). Thus, it's of vital significance to perform further research in this regard to validate and update the information acquired.

In the current study, we first screened a total of 252 (186 upregulated and 66 downregulated) DEGs from the unused GEO dataset GSE103512. The GO/KEGG pathway

enrichment analyses were implemented thereafter. Upregulated DEGs were mainly enriched in metabolic pathways, FA metabolism, and PPAR signaling pathway; whereas downregulated DEGs were mostly associated with protein digestion and absorption. Metabolism alteration is a hallmark of cancer. It is well demonstrated that the cancerous cellular metabolisms were urged to adapt themselves to the high proliferation rate and nutritional requirement so as to constantly support the increased cell division (*Chandel, 2014*; *Sancho, Barneda & Heeschen, 2016*). For PCa, one of the most noticeable metabolic changes is, as our results indicated, the FA or lipid metabolism (*Deep & Schlaepfer, 2016*; *Yang et al., 2016*). The lipid biosynthesis is essential for membrane formation and cell signaling; for instance, the metabolic intermediates of de novo lipogenesis can serve as second messengers and regulate PCa migration and invasion (*Ferro et al., 2017*). Moreover, the lipid metabolism of PCa is closely related to androgen by androgen receptor (AR) signaling. For examples, AR signaling can elevate the uptake of exogenous lipids by PCa cells (*Liu, Zuckier & Ghesani, 2010*) as well as stimulate adipose tissues to release FA (*O'Reilly, House & Tomlinson, 2014*). Furthermore, the PPARs family participate participates greatly in metabolic regulation. For example, PPARγ is a nuclear FA receptor which can interact with androgen receptor (AR) and regulate growth of PCa (*Olokpa, Moss & Stewart, 2017*). The FA-binding protein 5 (FABP5)-PPARγ signaling pathway promotes the malignant progression of castration-resistant PCa cells (*Jing et al., 2000*; *Kawaguchi et al., 2016*). Recently, the FABP5 inhibitor dmrFABP5 were illustrated to inhibit the FABP5-PPARγ pathway to suppress the malignant progression of PCa cells (*Al-Jameel et al., 2019*). Therefore, targeting the above pathways might aid in developing effective strategy for the treatment of PCa.

Next, we constructed PPI networks with the DEGs and identified the following top five hub genes: KLK3, CDH1, KLK2, FOXA1, and EPCAM. All hub genes exhibited upregulated expressions in PCa compared to NP tissues. Both KLK3 and KLK2 belong to the Kallikrein-related peptidases (KLKs) family. Expressed and secreted by glandular epithelia of many organs including skin, mammary gland, prostate, colon, pancreas and brain, KLKs are a sort of serine proteases existed in the body fluids excreted by the above organs such as sweat, milk, and seminal fluid, involving in various physiological functions like electrolyte balance, extracellular matrix remodeling, and prohormone processing (*Borgono, Michael & Diamandis, 2004*; *Shaw & Diamandis, 2007*). Aberrant expression of KLKs can be found in several kinds of malignancies including ovarian cancer (*Loessner et al., 2018*), breast cancer (*Figueroa et al., 2018*), gastrointestinal cancer (*Kontos et al., 2013*), and PCa (*Mavridis, Avgeris & Scorilas, 2014*; *McDonald & Parsons, 2016*). Further on, it was suggested that KLKs could lead to proliferation of the epithelium via protease-activated receptors, which might contribute to PCa development (*Ramsay et al., 2008*). Several studies have highlighted the association between KLK3 gene polymorphism and susceptibility to PCa (*Chen & Xin, 2017*; *Ding et al., 2018*; *Motamedi et al., 2019*), and *Zambon et al. (2012)* successfully combined the KLK3 genetics analysis and free-to-total PSA ratio for PCa diagnosis. Similarly, KLK2 was reported to be potential marker for diagnosis because its polymorphism can increase vulnerability to PCa (*David et al., 2002*; *Nam et al., 2006*). And more notably, the loss of miR-378

(a regulator of KLK2) in PCa was found to be correlated with aggressive phenotype and short-term relapse events (*Avgeris, Stravodimos & Scorilas, 2014*). Both KLK2 and KLK3 were reported to elevate the level of insulin-like growth factor (IGF) by cleaving the IGF-binding proteins, which consequently affects cell survival, mitogenesis, and differentiation. Analogously, KLK3 can cleave the TGFβ-binding proteins and activate TGFβ, resulting in cell proliferation (*Borgono, Michael & Diamandis, 2004*).

The encoding product of CDH1 gene, namely E-cadherin, participates in cell-cell adhesion and the maintenance of cell differentiation as well as normal structure of epithelium (*Mayer et al., 1993*; *Takeichi, 1995*). Pathologically, inactivation or absence of CDH1 expression can decrease intercellular junctions and thus promote cancer invasion and metastasis, as seen in diffuse gastric cancer (*Melo et al., 2017*) and lobular breast cancer (*McVeigh et al., 2014*). Contrary to expectation, our results indicated that CDH1 expression turned out to be significantly upregulated in PCa compared to NP tissues. Consistent with our results, another study using the GSE26910 dataset also found overexpressed CDH1 in PCa samples (*Fang et al., 2017*). It was hypothesized that overexpression of CDH1 might have an impact on oncogenesis since previous research has indicated the intercellular adhesion between cancer and normal cells were enhanced in the late stage of tumor formation (*Albelda, 1993*). Plus, polymorphism of CDH1 was also found to elevate the risk of PCa (*Imtiaz et al., 2019*; *Qiu et al., 2009*). Future work is merited to explore the expression changes of CDH1 in different stages of PCa.

Forkhead Box A1, also known as hepatocyte nuclear factor 3α, is a transcription factor that plays important roles in development as well as cancer formation (*Bernardo & Keri, 2012*). FOXA1 has multiple impact on PCa, including (1) controls the morphogenesis and cell differentiation of prostate (*Gao et al., 2005*); (2) facilitates AR transactivation which is essential for PCa proliferation and survival (*Zhao, Tindall & Huang, 2014*); (3) inhibits PCa progress towards neuroendocrine prostate cancer, whose prognosis is worse (*Kim et al., 2017*); (4) downregulates TGF-β signaling and thus suppresses castration-resistant prostate cancer progression (*Song et al., 2019a*). Our findings were validated by another expression profiling study (*Endo et al., 2009*), and the complexity of functions of FOXA1 still calls for further research to fully interpret its role in PCa development and progression.

Epithelial cell adhesion molecule encodes epithelial cell adhesion molecule, also known as CD326, is usually high expressed in cancer tissues and participates in intercellular adhesion limitation, cell signaling, migration, proliferation, and differentiation (*Maetzel et al., 2009*). Consistent with our results, EPCAM is shown to be overexpressed in localized and metastatic PCa (*Massoner et al., 2014*). Other integrated bioinformatic analysis using four GEO datasets also validated our findings (*Lu & Ding, 2019*).

When it comes to validation of the identified hub genes, three of them (CDH1, FOXA1, and EPCAM) were validated by other studies using relevant GEO datasets; four genes (except KLK2) were validated by TCGA data; and all five genes were validated by the Oncomine data. This implied their potentials as effective and reliable biomarkers for diagnosis and as possible therapeutic targets. However, in survival analysis using TCGA cohorts, all hub genes turned out to have insignificant impact on PCa patients' OS. To add

readability, we put the top 10 upregulated and downregulated DEGs into the survival analysis, outputting two significant genes, namely ABCC4 and SLPI. ABCC4 (ATP-cassette binding protein 4), which was significantly upregulated as our results revealed, its relatively higher expression was associated with poorer OS of PCa patients. This negative effect of ABCC4 expression toward PCa might result from that ABCC4 could decrease the efficacy of docetaxel in treating PCa cells (Oprea-Lager et al., 2013). Conversely, the relatively higher level of downregulated DEG SLPI (Stomatin-like protein 1) was related to better OS of PCa patients. There are not many studies focused on SLPI. Intriguingly, it is revealed that overexpression of SLPI stabilizes F-box protein Fbw7-γ by inhibiting its degradation, which is implicated in the degradation of oncogene c-Myc (Zhang, MacDonald & Koepp, 2012). So SLPI might impact negatively on cell proliferation of PCa through the above mechanism. Therefore, these two genes might hopefully serve as prognostic indicators for PCa patients.

We also conducted a module analysis on the existing PPI network and selected the top two significant modules for the ensuing KEGG pathway enrichment analysis of the genes contained. The results demonstrated that the genes in Module 1 are mainly enriched in adherens junction and Hippo signaling pathway; while the genes in Module 2 are mainly enriched in FA metabolism, fatty acid biosynthesis, PPAR signaling pathway, and FA degradation. Apparently, the modules KEGG analyses were quite conformed to the overall KEGG results. The FA metabolism related pathways, PPAR signaling pathway, and cell adhesion-related hub genes and pathways were already discussed above. It was demonstrated that the activation of Hippo signaling pathway regulates cell growth and proliferation by inhibiting YAP and TAZ transcription co-activators, and its dysregulation impacts significantly on development of cancers including colon, liver, breast, lung, ovary, and prostate (Salem & Hansen, 2019; Yu, Zhao & Guan, 2015).

## CONCLUSIONS

By bioinformatic analyses including GO/KEGG enrichment, PPI network, hub gene identification, and module analysis, the current study validated KLK3, CDHI, FOXA1, and EPCAM might potentially serve as effective and reliable molecular biomarkers for diagnosis of PCa; and ABCC4 and SLPI might be utilized as prognostic indicators of PCa. However, further basic and clinical researches are necessary for the verification of the clinical value of our findings.

## ACKNOWLEDGEMENTS

Special thanks to Mr. Fucai Tang for his assistance in the use of R programing Language.

### Funding

This work was supported by the Guangdong Provincial Science and Technology Plan Project (No. 2017B030314108). The funders had no role in study design, data collection and analysis, decision to publish, or preparation of the manuscript.

## Grant Disclosures

The following grant information was disclosed by the authors:
Guangdong Provincial Science and Technology Plan: 2017B030314108.

## Competing Interests

The authors declare that they have no competing interests.

## Author Contributions

- Zihao He conceived and designed the experiments, performed the experiments, analyzed the data, prepared figures and/or tables, authored or reviewed drafts of the paper, approved the final draft.
- Xiaolu Duan conceived and designed the experiments, contributed reagents/materials/analysis tools, authored or reviewed drafts of the paper, approved the final draft.
- Guohua Zeng conceived and designed the experiments, contributed reagents/materials/analysis tools, authored or reviewed drafts of the paper, approved the final draft.

## Data Availability

Data is available at NCBI GEO: GSE103512.

## Supplemental Information

Supplemental information for this article can be found online at http://dx.doi.org/10.7717/peerj.7872#supplemental-information.

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
