# Peer review of "Identification of potential biomarkers and pivotal biological pathways for prostate cancer using bioinformatics analysis methods"

_PeerJ, doi:10.7717/peerj.7872_

## Round 0.1 · original submission · Major Revisions

Both the reviewers have suggested major changes. Please incorporate all the suggested changes and resubmit the manuscript.

Reviewer 1 ·

Basic reporting

In this work, the authors used bioinformatics tools to analyze the gene expression profile of prostrate cancer from GEO database and identified key genes in the process that may serve as molecular biomarker.

The manuscript lacks on sufficient background information and literature references as further elaborated in general comments.

Experimental design

The aim of the study is well-defined and the methods used are appropriate and sufficiently detailed.

Validity of the findings

Conclusions are well stated but validity of the findings not sufficiently assessed as also noted in general comments.

Additional comments

1) Introduction needs improvement and should be rewritten as sufficient background covering previous bioinformatics work is not provided to understand the significance and novelty of the work. For example, lines 57-59, “However, the underlying molecular mechanisms and hub genes or key signaling pathways of PCa were still far from explicit”. There have been several bioinformatics work identifying hub genes in PCa (to mention a few: PMIDs: 29260398, 28800317, 30405806 etc.). Authors should provide information on key genes identified in previous studies, and in the discussion elaborate on the similarities/differences in the hub genes identified in this manuscript to those already reported in literature.
2) In Table 1, it seems that the top 10 upregulated and downregulated DEGs are identified by their P-values which do not necessarily represent the top regulated genes by fold-change (as can be seen in Figure 1 which identifies genes with fold change values up to ~3). Can authors comment on why fold-change was not used as the criteria here.
3) Authors should explain in the manuscript why GSE103512 dataset was picked in particular for the analysis since there are other PCa datasets available too (for example: GSE55945). More importantly, authors should validate the hub genes identified using GSE103512 against other PCa datasets.
4) There are spelling mistakes at several places in the manuscript. For example, “defferentially expressed genes” on line 63.

Reviewer 2 ·

Basic reporting

Clear and unambiguous, professional English used throughout.

Experimental design

Methods described with sufficient detail & information to replicate.

Validity of the findings

Conclusions are well stated, linked to original research question & limited to supporting results.

Additional comments

I thank the editor for providing an opportunity to review the article “Identification of Potential Biomarkers and Pivotal Biological Pathways for Prostate Cancer Using Bioinformatics Analysis Methods”.

The article is written well and can be considered for publication after addressing these comments:
1. Fig 1, some of the important genes name should be labeled (especially genes with very high changes).
2. Figures should be combined together, and a total number of figures should be less (e.g. combine Fig 1 and 2).
3. Pathway analysis and conclusions are very vague. Cannot be used for therapeutic purpose. Please discuss.
4. Module analysis showing enriched interaction between many genes, specially KLK3, CDH1, KLK2, FOXA1, and EPCAM. These genes should be annotated and highlighted in Fig 1, 2.
5. This study is based on one library. The author should validate their findings from other available library and TCGA database.

---

## Round 0.2 · Minor Revisions

Please address the critical query raised by Reviewer 2.

Reviewer 1 ·

Basic reporting

NA

Experimental design

NA

Validity of the findings

NA

Additional comments

Authors have addressed all my concerns and the manuscript is now suitable for publication.

Reviewer 2 ·

Basic reporting

N/A

Experimental design

N/A

Validity of the findings

N/A

Additional comments

Authors have addressed most of the suggestion and comments. This article can be considered for publication after addressing comment:

1. How come upregulated and downregulated genes are flipped in the revised version? And also, for pathway analysis? If the analysis was done in a similar way, how come results are opposite? Please discuss and clarify.

---

## Round 0.3 · accepted · Accept

Authors have incorporated all the suggested changes. The manuscript is ready for publication.

Reviewer 2 ·

Basic reporting

N/A

Experimental design

N/A

Validity of the findings

N/A

Additional comments

Authors have addressed all suggestion and comments. I recommend this article for publication.